# A Multi-sensor Data-driven Methodology for All-sky Passive Microwave Inundation Retrieval

Zeinab Takbiri[1], Ardeshir M Ebtehaj[1], Efi Foufoula-Georgiou[3,1]

[1] Department of Civil, Environmental and Geo- Engineering and St. Anthony Falls Laboratory, University of Minnesota, Twin Cities, Minneapolis, USA.

[3] Department of Civil and Environmental Engineering, University of California, Irvine, USA

*Correspondence to*: Zeinab Takbiri (takbi001@umn.edu)

**Abstract.** We present a multi-sensor Bayesian passive microwave retrieval algorithm for flood inundation mapping at high spatial and temporal resolutions. The algorithm takes advantage of observations from multiple sensors in optical, short-infrared, and microwave bands, thereby allowing detection and mapping of the sub-pixel fraction of inundated areas under almost all-sky conditions. The method relies on a nearest neighbor search and a modern sparsity-promoting inversion method that make use of an *a priori* dataset in the form of two joint dictionaries. These dictionaries contain almost overlapping observations by the Special Sensor Microwave Imager and Sounder (SSMIS) on board the Defense Meteorological Satellite Program (DMSP) F17 satellite and the Moderate Resolution Imaging Spectroradiometer (MODIS) on board the Aqua and Terra satellites. Evaluation of the retrieval algorithm over the Mekong delta shows that it is capable of capturing to a good degree the inundation diurnal variability due to localized convective precipitation. At longer time-scales, the results demonstrate consistency with the ground-based water level observations, denoting that the method is properly capturing inundation seasonal patterns in response to regional monsoonal rain. The calculated Euclidean distance, rank-correlation and also Copula quantile analysis demonstrate a good agreement between the outputs of the algorithm and the observed water levels at monthly and daily time scales. The current inundation products are at resolution of 12.5 Km and twice per day, but higher resolution (order of 5 Km and every 3 hours) can be achieved using the same algorithm but populating the dictionary with the Global Precipitation Mission (GPM) Microwave Imager (GMI) products.

Keywords:

Passive Microwave Inundation Retrievals, Bayesian Inversion, *k*-nearest Neighbors, Deltaic regions, Inverse Problems, Sparse Regularization.

Key points:

Multi-sensor observations improve satellite inundation mapping under cloudy sky.

Retrievals using passive microwave observations capture the diurnal variability of inundation.

## 1 Introduction

Capturing the diurnal spatio-temporal dynamics of inundation over coastal regions, deltaic surfaces, and river floodplains requires high-resolution observations in both time and space, which are not available from the typical sparse ground-based sensors. Satellite observations from the visible to the microwave bands of the electromagnetic spectrum have been widely used for mapping floods, estimating surface water storages, river discharge values and water levels (Smith, 1997). In the visible bands (~0.4–0.8 µm), natural water reflects a fraction of incident light depending on the water depth and concentration of the optically active components such as suspended and dissolved particulate matter. However, water reflectivity sharply declines and approaches zero in the near infrared bands (~0.8–2.5 µm). Thresholding of this sharp gradient is often used to discriminate water bodies from their nearby dry soils and vegetated surfaces (Rango and Anderson, 1974; Smith, 1997 and references therein; Frazier and Page, 2000; Smith, 2001; Jain et al., 2005). In the microwave region of the spectrum, the dielectric constant of water (~80) is much higher than the dry soil (~4) and thus the inundated areas are substantially less emissive and radiometrically colder than the surrounding soils and vegetation covers. Moreover, emission from smooth water surfaces is more polarized than that from rough soils and vegetated surfaces (Ulaby et al., 1982; Papa et al., 2006; Prigent et al., 2007). This polarization signal has also been used through empirical thresholding approaches to distinguish water surfaces from other land surface types (Allison et al., 1979; Sippel et al., 1994, 1998; Brakenridge et al., 2005, 2007).

Flood mapping from space was first accomplished using visible to near infrared (VNIR) observations (0.4–1.1µm), by the Multispectral Scanner System (MSS) sensors on board Landsat-1 (Rango and Anderson, 1974; Rango and Salmonson, 1974; McGinnis and Rango, 1975). In these pioneering works, flooded areas were mapped where the near-infrared surface reflectance was below a certain threshold as water absorption is strong in this region. More recently, Brakenridge and Anderson (2006) showed that the visible red band 1 (0.62–0.67µm) and near infrared (NIR) band 2 (0.84–0.87µm) from the Moderate Resolution Imaging Spectroradiometer (MODIS) aboard the Terra and Aqua satellites can be used to detect water over land surfaces. They mapped several hundreds of flood events at different sites all over the world by classification of water via thresholding over the NIR band and the normalized difference vegetation index, $NDVI = (NIR - red)/(NIR + red)$ introduced by Rouse et al. (1974). To better discriminate the vegetation from inundated areas in threshold-based methods, Ticehurst et al. (2013) and Guerschman et al. (2011) used a new index—called the normalized difference water index, $NDWI = (red - MIR)/(red + MIR)$ introduced by Gao (1996) and later modified to $MNDWI = (green - MIR)/(green + MIR)$ by Xu (2006). This index exploits the mid-infrared (MIR: 1.23–1.25µm) part of the spectrum to improve the mapping. In all thresholding methods, the shadows of terrains and clouds are usually miss-classified as inundated areas. Therefore, Kuenzer et al. (2015) used the topography and cloud information data as ancillary variables to obtain improved estimates of the inter-annual dynamics of areas covered with water over five deltaic regions with high annual cloud cover.

The use of passive microwaves (PMW) to map flooded areas was pioneered by Allison et al. (1979), Giddings and Choudhury (1989), and Choudhury (1991). Allison et al. (1979) used horizontal polarization of brightness temperatures (Tb) at 19.3 GHz, from the Electrically Scanning Microwave Radiometer (ESMR) on board the Nimbus-5 satellite, to delineate flooded regions in Australia. Giddings and Choudhury (1989) reported the 37GHz vertical and horizontal polarization differences (i.e.,

$Tb_{37v} - Tb_{37h}$ ), from the Scanning Multi-frequency Microwave Radiometer (SMMR) on board the Nimbus-7 satellite, as the most responsive channel to identify the seasonal changes in the extent of floodplains over South America. Temimi et al. (2005) used the empirical Basin Wetness Index (BWI) defined by Basist et al. (1998), to obtain real-time water surface fraction (WSF) in the Mackenzie River Basin, using multi-frequency information at 19, 37, and 85 GHz. To minimize the contamination effects of atmospheric emission and variations of surface temperatures, Brakenridge et al. (2007) exploited the ratio of Tb

values over inundated and dry surfaces at 36 GHz and presented promising results over several river sites all over the globe, using the PMW observations by the Advanced Microwave Scanning Radiometer - Earth Observing System (AMSR-E). De Groeve et al. (2010) also used the same method and instrument to map floods for several hundreds of locations for the Global Disaster Alert and Coordination System GDACS.

While visible and shortwave-infrared bands often provide sub-kilometer resolution for inundation mapping, their capability is very limited in a cloudy sky. This limitation is usually very restrictive over prone-to-flooding watersheds and deltas in tropical regions with high-frequency of heavy precipitation events. For instance, a long-term analysis of Landsat data revealed that due to cloud contamination, only 30% of overpasses are useful for inundation mapping (Melack et al., 1994). Because of this limitation, most of the related satellite products, including the MODIS inundation products, are available mostly in monthly,

seasonal, and/or annual timescales (Ordoyne and Friedl, 2008). However, microwaves can penetrate clouds—and to some extent hydrometeors in frequencies $\leq 37$ GHz—to provide water inundation mapping in almost all weather conditions. Unfortunately, due to the coarse resolution of microwave data, e.g., $(47 \times 74)$ km$^2$ at 19 GHz to $(13 \times 16)$ km$^2$ at 183 GHz for the SSMIS), only large water bodies can be detected and sub-pixel inundated areas cannot be directly identified (Smith, 1997). Nowadays, there exist several sensors on board different satellites which overlap in the spatial and time domains that sample

land-atmosphere signals at different wavelengths of the electromagnetic spectrum. Therefore, it is imperative to integrate these multi-sensor observations to overcome their individual shortcomings and improve retrievals of land-atmosphere parameters and the extent of flooded areas (Prigent et al., 2001, 2007; Crétaux et al., 2011; Temimi et al., 2011; Schroeder et al., 2010).

In this paper, we develop a method to retrieve sub-pixel inundation fraction ("inundation" referring to regions where water

covers the land surface, excluding permanent water bodies) only from passive microwave observations based on a set of paired VNIR and passive microwave training samples. In particular, as training samples, we use global observations of VNIR data from the Moderate Resolution Imaging Spectroradiometer (MODIS) on board Terra (launched in 2000) and Aqua satellites (launched in 2002) and passive microwave data from the Special Sensor Microwave Imager/Sounder (SSMIS) on board DMSP

satellites F16–F18. Several years of observations (2000-present) by these two sensors allow us to collect adequate overlapping data to link coarse scale SSMIS passive microwave data to high-resolution MODIS VNIR data in the form of organized dataset. Obviously, this collection of almost coincident observations does not contain direct information about surface inundation in a cloudy sky, as the radiative signals in VNIR wavelengths cannot penetrate clouds. However, over land, it is well understood

(see Ferraro et al., 1986; Grody, 1991; Wilheit, 1994) that hydrometeors and the atmospheric profile do not significantly affect the low-frequency <60 GHz brightness temperatures. Therefore, the information content of the dataset over low-frequency channels is independent of the atmospheric profile and can be used to a good degree of accuracy to recover inundated surfaces under cloudy conditions as well. It should be acknowledged that there is an uncertainty for the inundation retrieval under heavy rainy/cloudy sky when only the information in the clear-sky dataset is used. However, we expect that this uncertainty will be

small since the information of the underlying surfaces in low-frequency channels of the collected dataset remains almost the same over different atmospheric conditions.

The collected dataset has a large number of linked pairs of inundation fraction data from MODIS data SSMIS multi-frequency brightness temperatures. For algorithmic development, the dataset is organized into two fat matrices: the so-called brightness

temperature and inundation dictionaries. For an observed pixel-level brightness temperature, the proposed passive retrieval algorithm uses the nearest-neighbor search to isolate a few vectors in the dictionary of brightness temperatures and their corresponding inundation fraction and then use them to estimate the unknown inundation fraction. The proposed retrieval algorithm is applied to estimate daily inundation fraction at spatial resolution of 12.5 km over the Mekong in 2015. The main motivation for selecting this delta as a case study is that approximately 90% of the Mekong region is covered by clouds during

the rainy season (Leinenkugel et al. 2013) which severely hampers the use of inundation mapping in the VNIR bands. We retrieve the inundation fraction twice per day using the proposed algorithm over the Mekong delta and compare the results with the flood products of VNIR data during clear skies. We also evaluated the results against the daily and monthly water level data obtained from eleven gages over the Mekong delta (Fig. 1) to examine consistency of the retrievals with the regional inundation patterns.

This paper is organized as follows. Section 2 explains the *a priori* dataset and the formation of the dictionaries and Section 3 provides detailed information about the retrieval algorithm. Implementation of the method and validation are explained in Section 4. Section 5 presents concluding remarks and directions for future research.

## 2 Study Area and dataset

The 60,000-km$^2$ Mekong delta is in South Vietnam (see Fig. 1) with a tropical monsoon climate system. The delta with its agricultural industry is one of the most important sources of food supply to the Southeast Asia. This critical region is home to nearly 20 million people, approximately 22% of the population of Vietnam, and is one of the most densely populated regions

in the world. The area has been exposed to exacerbated erosion due to human activities and increased sea level rise and lowland flood events in the recent decades (e.g., Syvitski et al., 2005; Ericson et al., 2006; Nicholls and Cazenave, 2010; Tessler et al., 2015). Improved quantification of (near) real-time inundation of the Mekong Delta can help: (1) to improve flood forecasting by identifying the inundated and thus soil saturated zones and (2) to identify erosional and depositional hotspots that can improve geomorphologic and ecosystem modeling. The proposed retrieval algorithm is applied to estimate sub-daily inundation fraction at resolution of 12.5 km over some of the lower regions of the Mekong delta in calendar year 2015 (Fig. 1).

Two sources of information are used to build a dataset that connects almost coincident VNIR water inundation data and multi-frequency passive microwave data. The VNIR data consist of the daily NASA standard MODIS Near-Real-Time (NRT) Water Product (MWP-3D3ON i.e., 3-Days imagery, 3 Observations, and no shadow masking) with approximately 250 m spatial resolution (Nigro et al., 2014) from both Terra and Aqua Satellites. The Terra and Aqua satellites both have a sun-synchronous orbit. They rotate around the earth in opposite directions: Terra has an ascending orbit with the local equatorial crossing time of 10:30 AM and Aqua has a descending orbit with the local equatorial crossing time of 1:30 PM. MWP products are binary information of inundation based on the Dartmouth Flood Observatory (DFO) algorithm, which uses a thresholding scheme on MODIS observations at Band 1 (0.62–0.67 µm), Band 2 (0.84–0.87 µm) and Band 7 (2.10–2.15 µm). To minimize the contamination effects of cloud and terrain shadows, we focus on 3-day composite MWP products (3D3ON). Clearly, the use of the 3-day composite MODIS-MWP data can affect daily inundation retrievals; however, in the context of the presented algorithm this is the best choice because, daily MODIS-MWP composites are very uncertain due to the terrain shadows and clouds (Nigro et al. 2014). Typically, there are numerous missing pixels in the daily products, which reduce the sample size dramatically. These errors are significantly reduced in 3-day composite products, as it is less likely that clouds (and their shadows) stay at the same spot during a 3 day period (Nigro et al. 2014).

The microwave data are obtained from the DMSP SSM/I-SSMIS Pathfinder Daily Equal-Area Scalable Earth Grid (EASE-Grid; see Armstrong and Brodzik, 1995) brightness temperatures distributed by the National Snow and Ice Data Center (NSIDC). These datasets are at four central frequencies 19, 22, 37, and 91 GHz. All channels are vertically and horizontally polarized except channel 22 GHz. The effective resolution of the highest frequency channel is ~12.5 km while low-resolution channels are projected onto a grid size of ~25 km. DMSP SSM/I-SSMIS brightness temperature data products are from observations by the SSM/I and SSMIS radiometer on board the DMSP F8, 11, 13 or 17. Since December 2006, the F17 satellite has been the only operational satellite from the DMSP series, which carries on board the SSMIS instrument with equatorial crossing times of 05:30–06:30 AM and 17:30–18:30 PM for the descending and ascending orbits, respectively. It is important to note that because these satellites revisit every point on earth at the same local time, repeatedly, the paired MODIS-MWP with DMSP SSMIS data have a fixed diurnal time-difference in the entire dataset. Since the MODIS-MWP data are from the

combination of Terra and Aqua observations, their time tag is advantageous in the sense that it allows us to enrich the number of samples for the diurnal cycle of inundation dynamics.

The first step for building the *a priori* dataset is to match the different space-time resolutions of the multi-sensor information. To unify the spatial resolution of the microwave data, the brightness temperatures of the three lower frequency channels are
mapped onto the latitude/longitude grids of the high-frequency channel of 91 GHz with resolution ~12.5 km, using a nearest neighbor interpolation. Then the clear-sky MWP data are also upscaled from 250 m to 12.5 km and projected onto the same grids. In the process of upscaling the binary MWP data, we assigned to each upscaled pixel a scalar inundation fraction number $f$ that represents the ratio of the number of inundated sub-pixels to the total number of sub-pixels within a pixel size of 12.5 km. For matching the time scales of Tb and MWP values, the Tb values are averaged over a three-day time window to minimize
the possible effects of cloud contamination in the VNIR data. Fig. 2 demonstrates schematically the process of producing the explained dataset.

## 3 The Retrieval Algorithm

The proposed retrieval algorithm uses the link between two available coincidental datasets, passive microwave (SSMIS) and VNIR (MODIS-MWP), to retrieve inundation in the cloudy days. First, the overlapped clear-sky pixels of MODIS-MWP and
SSMIS for 5 years (2010–2014) are collected over the study area to create two coincidental dictionaries: the SSMIS dictionary and the MODIS-MWP dictionary. The SSMIS dictionary consists of 8-dimensional vectors of brightness temperature (Tb), where 8 is the number of frequency channels, and the MODIS-MWP dictionary consists of scalar values of inundation fractions for each corresponding pixel in Tb. In other words, the inundation fraction for each Tb in the brightness temperature dictionary is known. The algorithm uses the information embedded in these two dictionaries to estimate the unknown inundation fractions
for each Tb observation vector. First, it searches the brightness temperature dictionary to find the *K* most similar vectors in the Euclidean sense to the Tb observation vector through the *K*-nearest neighbors algorithm. Then, for these *K*-nearest neighbors, the corresponding known scalar values in the inundation fraction dictionary are picked. If the ratio of the number of inundated vectors in *K*-nearest neighbors is greater than a threshold (which will be explained later), this pixel is called inundated and the algorithm goes to the estimation step. In the estimation step, the coefficients that can optimally estimate the Tb observation
vector based on its *K*-nearest neighbors are calculated through a least squares regularization approach. Those coefficients are then used to linearly combine the *K* known inundation fractions that are associated with the neighboring Tb vectors for calculating the unknown inundation fraction. The above detection and estimation steps are repeated for each orbit at pixel-level of 12.5 km over the study area. The algorithm is mathematically described in what follows.

To organize the dataset in an algebraically tractable manner, *M* vectors of microwave brightness temperatures $\mathbf{b}_i = \left( Tb_{1i}, Tb_{2i}, ..., Tb_{ni} \right)^T \in \Re^n$ at *n* frequency channels are collected. These vectors form the column space of an *n*-by-*M* matrix

$\mathbf{B} = \begin{bmatrix} \mathbf{b}_1 \mid \mathbf{b}_2 \mid ... \mid \mathbf{b}_M \end{bmatrix} \in \mathfrak{R}^{n \times M}$, called brightness temperature dictionary, where $M \gg n$. Analogously, the corresponding inundation fraction values $\{f_i\}_{i=1}^{M}$ can be collected in the column space of the inundation dictionary $\mathbf{F} = \begin{bmatrix} f_1 \mid f_2 \mid ... \mid f_M \end{bmatrix} \in \mathfrak{R}^{1 \times M}$ For each vector, $\mathbf{b}_i$ in the dictionary of brightness temperatures there is an inundation fraction $f_i$ from MODIS-MWP. The collection of these pairs from historical observations forms the two dictionaries $\mathbf{B}$ and $\mathbf{F}$. The algorithm follows two sequential steps: a detection and an estimation step. In the detection step, for each observed vector of brightness temperature $\mathbf{b}_{\mathrm{obs}}$, the algorithm first finds its $K$-neighboring brightness temperatures in $\mathbf{B}$ in the Euclidean sense and stores them in the column space of $\mathbf{B}_s \in \mathfrak{R}^{n \times K}$. Then, knowing the column indices of the neighboring brightness temperatures, it isolates their corresponding inundation fraction values in $\mathbf{F}_s \in \mathfrak{R}^{1 \times K}$. In this step, if at least $p \times K$ number of nearby inundation fraction values in $\mathbf{F}_s$ are non-zero, the algorithm assumes that $\mathbf{b}_{\mathrm{obs}}$ is over an inundated pixel and attempts to estimate the fraction of inundation in the estimation step. Here, $p \in (0-1)$ is the detection probability parameter. It should be also noted that the $K$-nearest neighbor algorithm in this paper does not directly constrain its search to any specific time or location. In other words, for every pixel-level vector of Tb, the $K$-nearest neighbors algorithm searches the entire dictionary regardless of any specific time or spatial coherency.

In the estimation step, the method assumes that $\mathbf{b}_{\mathrm{obs}}$ can be estimated by a linear combination of a few column vectors of $\mathbf{B}_s$ as follows:

$$\mathbf{b}_{\mathrm{obs}} = \mathbf{B}_s \mathbf{c} + \mathbf{e} \tag{1}$$

where the vector $\mathbf{c} \in \mathfrak{R}^K$ contains a set of representation coefficients to be estimated and $\mathbf{e} \in \mathfrak{R}^n$ is the error vector. Clearly, for an observed vector of brightness temperatures $\mathbf{b}_{\mathrm{obs}}$, the goal is to estimate its unknown inundation fraction value $\hat{f}$. We assume that the two paired dictionaries $\mathbf{B}_s$ and $\mathbf{F}_s$ represent similar manifolds in a geometric sense that their local structures can be approximated well with the same linear model. This allows us to assume that the representation coefficients in vector $\mathbf{c}$ from Eq. (1) can be used to estimate the inundation fraction $\hat{f}$ as follows:

$$\hat{f} = \mathbf{F}_s \mathbf{c} \tag{2}$$

As a result, using a classic weighted least-squares method, the representation coefficients $\mathbf{c}$ can be estimated as:

$$\hat{\mathbf{c}} = \underset{\mathbf{c}}{\mathrm{argmin}} \left\{ \left\| \mathbf{W} \left( \mathbf{b}_{\mathrm{obs}} - \mathbf{B}_s \mathbf{c} \right) \right\|_2^2 \right\} \tag{3}$$

where $\mathbf{W}$ is a weight matrix (to be discussed later in this section) that characterizes the importance of each channel in the retrieval scheme. The number of $K$-nearest neighbors is often larger than the number of frequency channels, $k \gg n$, making

$\mathbf{B}_s$ a rank-deficient matrix and the above problem ill-posed. To make the optimization problem (3) well-posed, we use a mixed $\ell_1 - \ell_2$ norm regularization as follows:

$$\hat{\mathbf{c}} = \underset{\mathbf{c}}{\text{Argmin}} \ \left\{ \left\| \mathbf{W}\left(\mathbf{b}_{\text{obs}} - \mathbf{B_s}\mathbf{c}\right) \right\|_2^2 + \lambda_1 \left\| \mathbf{c} \right\|_1 + \lambda_2 \left\| \mathbf{c} \right\|_2^2 \right\}$$
$$\text{subject to } \mathbf{c} \succeq 0, \ \mathbf{1}^{\text{T}}\mathbf{c} = 1 \tag{4}$$

which has been successfully used for passive microwave precipitation retrievals (Ebtehaj et al., 2015a, 2015b). The non-negativity of the coefficients assures positivity of the brightness temperatures and the sum-to-one constraint enforces an

5    unbiased estimation. The regularization involves both the $\ell_1$-norm $\left\| \mathbf{c} \right\|_1 = \sum_{i=1}^{K} \left| c_i \right|$ and the $\ell_2$-norm $\left\| \mathbf{c} \right\|_2 = (\sum_{i=1}^{K} \left| c_i \right|^2)^{\frac{1}{2}}$. The

parameters $\lambda_1$ and $\lambda_2$ in Equation (4) are regularization parameters that enforce a trade-off between the two regularizations $\ell_1$ and $\ell_2$. In this mixed regularization, the $\ell_1$-norm leverages sparsity in the solution (i.e., forces some of the elements of $\mathbf{c}$ to be zero) while the $\ell_2$-norm increases the stability of the solution as the neighboring brightness temperatures in $\mathbf{B}_S$ are likely to be highly correlated (see Zou and Hastie, 2005). In effect, due to the use of a mixed regularization, this regularization

10    promotes group sparsity (i.e., some blocks of the representation coefficients are zero) while it keeps the solution sufficiently stable. In other words, it acknowledges the fact that there are a few clusters of nearby brightness temperatures that can properly explain the observation. By enforcing the $\ell_1$-norm we select vectors that are parts of clusters of brightness temperatures, while the $\ell_2$-norm handles the potential correlation between those clustered neighbors and makes the problem sufficiently stable. The proposed algorithm is summarized in a flowchart shown in Fig. 3.

As previously noted, in the current implementation of the proposed retrieval algorithm, we focus on (almost) coincidental observations of the brightness temperatures and inundation fractions by the SSMIS and MODIS instruments, respectively. The dictionaries $\mathbf{B}$ and $\mathbf{F}$ are constructed using 5 years of overlapping data (2010–2014) over the Mekong delta (latitude: 0–10 N and longitude: 100–110 E) at 12.5 km grid resolution (Fig. 1).

To build the dictionary, only the clear-sky MODIS-MWP products were considered. At resolution 12.5 km, we labeled a pixel as clear-sky when less than 50 percent of the VNIR data at resolution 250m is flagged as non-cloudy. Because the MODIS sensor has a much higher resolution than the footprint of SSMIS and because the number of cloud-free samples over the Mekong is very limited, a threshold above zero is deployed to keep a certain number of partially cloudy pixels and make sure

25    that the dictionary will not be undersampled. For choosing the threshold, we conducted some sensitivity analysis (not shown here) and found a 50 percent threshold, as a fair probability choice, results in minimum of potential biases.

Since the DMSP satellites have two different equatorial crossing times, here, we use two sets of dictionaries for Tb values in the ascending (day or morning) and descending (night or evening) orbits. From all the available coincident observations, we randomly chose $2\times10^6$ pairs of brightness temperatures and inundation fractions in each ascending and descending dictionary. The purpose of stratifying the dictionaries into ascending and descending orbits is to exclude the effects of Tb modulations from the retrieval process caused by the systematic diurnal variation of surface temperature. In other words, the same inundation fraction has different PMW spectral signature in a daytime versus a night-time overpass largely due to the diurnal variability of skin temperature, precipitation, and soil moisture (see Mears et al., 2002; Ramage and Isacks, 2003; Norouzi et al., 2012). Fig. 4(a) presents the systematic difference between the Tbs of the ascending versus descending tracks for various ranges of pixel-level inundated fractions. In effect, in this figure, the Tbs in the dictionaries are grouped into five intervals based on their corresponding inundation fraction (from 0 to 1) in $\mathbf{F}$. Then for each interval, the average of Tb values is shown. The plot clearly demonstrates that the daytime Tbs are thermally warmer than their night time counterparts and this difference begins to shrink when the inundation fraction increases. It is worth noting that the difference between ascending and descending brightness temperatures is larger over the low-frequency channels ($\leq 37$ GHz) as they respond more to the land surface structural variability than the higher frequency channels that capture atmospheric signatures. Fig. 4(b) depicts $\left|Tb_A - Tb_D\right|$ where $Tb_A$ and $Tb_D$ stands for ascending and descending overpasses, respectively. It can be observed that high values of $\left|Tb_A - Tb_D\right|$ depict the coastlines, i.e., regions with the transient presence and/or absence of water over land.

The probability of detection, $p \in (0-1)$, determines if a pixel is inundated or not if the number of inundated vectors in K-nearest neighbors is $\geq p \times K$. We found that the inundation detection with $K \geq 50$ gives a reasonable rate for the probability of hit and false alarm. In other words, the probability of detection does not change significantly for a larger number of nearest neighbors. In the estimation step, to characterize the weight matrix $\mathbf{W} \in \Re^{n\times n}$, we used the coefficients of variation of each channel in response to changes in the inundation fraction (see Fig. 5). In other words, we assume that those channels that exhibit more variability with respect to changes in inundation fraction contain more information about inundation and shall be given more weight in the estimation process. One might ask why it is important to consider the high-frequency channels (e.g., 91 V, H GHz) despite the fact that they show minimal sensitivity to the inundation fraction (Fig. 5) and land surface emissivity compared to lower frequency channels. The high-frequency channels mainly capture the information content of the atmospheric profile. Therefore, incorporating them in the proposed retrieval framework allows us to indirectly consider the effect of atmospheric conditions by narrowing down the search for $K$-nearest neighbors to those Tb candidates that best match both the underlying land surface emissivity and the atmospheric conditions.

For implementation of the algorithm, the regularization parameters are set as $\lambda_1 = \lambda(1-\alpha)$ and $\lambda_2 = \alpha\,\lambda$, where $\alpha \in (0,1)$. Here, through cross validation studies, we empirically through cross-validation found that $\lambda = 0.001$ and $\alpha = 0.1$ provide a

reasonable balance between sparsity and stability of the solution in Eq. (4). It should be noted that Eq. (4) is converted to a constrained quadratic programming problem and solved using an iterative Newton's method with MATLAB optimization Toolbox (see Branch and Grace, 1996).

5    **4 Results, Validation and Discussion**

The inundation fractions were estimated during the wet period of calendar year 2015 from July-to-December when the water levels across the delta begin to rise and eventually recede (see Fig. 6). The wet season of the region is largely characterized by heavy precipitation as a result of the interactions of two monsoons including the Indian monsoon and the East Asia-Western North Pacific summer monsoon (Delgado et al., 2012).

To study the performance of the detection step we computed the probability of hit $P(\hat{f} > 0 | MWP > 0)$ and false alarm $P(\hat{f} > 0 | MWP = 0)$ of the algorithm outputs. Our analysis indicates that the probability of hit is around 0.92 for both the dry and wet season, demonstrating the capability of the algorithm in detecting the inundated areas. However, the probability of false alarm is around 0.12 for the dry season and reaches the value of 0.34 for the wet season, which might be due to the

15    generalization of the algorithm and MODIS missing data during the wet season. The MODIS daily data, especially in the wet season, contain a large number of missing values due to cloud blockages and frequent heavy rains over the study area. In fact, while we were collecting the overlapping data for constructing the dictionaries, we observed that over 88% of the MWP products have some missing portion in the 12.5 km resolution. As a result, it is very likely that the MWP data underestimate the actual inundation fraction of regions with prolonged precipitation events.

Fig. 6 shows that the algorithm is capable of identifying hotspots of inundation when its outputs are compared with the MODIS-MWP; however, the algorithm slightly overestimates the inundation fractions for some pixels farther from the coastlines, most of which are completely dry in MWP. Here for brevity, we only show the results for ascending overpasses, while similar spatial patterns are observed for descending overpasses. Fig. 6 also shows some overestimation of inundation fractions near the

25    riverbanks of major rivers. This might be due to the high soil moisture content ($\geq 0.8$) during the wet season that increases the dielectric constant of the soil up to 30-50 (Alharthi and Lange 1987), which is close to the dielectric constant of the water surfaces (75-80). Another reason, is the cloud coverage. Since the riverbanks are inundated less frequently than the coastlines, it is possible that these few inundation events were missed by MODIS because of the cloud blockages. There is also some underestimation in the inundation fractions from the proposed algorithm over the hillslopes far away from the riverbanks

30    compared to the MODIS-MWP product. Those sporadically inundated areas, which appear on the MODIS-MWP map (Fig 6. b & c), can be due to the terrain shadows that are misclassified as water. While we cannot directly prove the above assertion within the scope of this manuscript, the elevation map (Fig. 1) indicates that those hillslopes are very unlikely to get inundated.

Comparison of inundation fractions from MODIS-MWP and the proposed algorithm at daily scale is also challenging. This is because daily MODIS data are often severely corrupted by cloud coverage. On the other hand, under a clear sky, the MODIS-MWP inundation fractions are more precise than the results of the retrieval algorithm. To better illustrate this issue, scatterplots of daily inundation fractions from our retrieval algorithm against those from MODIS-MWP in wet and dry season are displayed in Fig. 7. The scatterplots further demonstrate larger inundation fractions from the retrieval algorithm in July-to-December (Fig 7(a)). In the wet season, there are also a lot of non-zero retrieved inundation values on the Y-axis that have corresponding zero inundated pixels in MODIS-MWP data. However, in January-to-June, when there are fewer clouds, the inundation fractions from the proposed algorithm are generally more correlated with the MODIS-MWP data but slightly underestimated (Fig. 7(b)). This underestimation probably also exists in wet months but is masked because of the all-sky retrieval capability of the proposed algorithm in the presence of the clouds and heavy rains. The reason for this underestimation might stem from the choice of 50 percent threshold for selecting the clear-sky pixels. In other words, there are a set of brightness temperatures for which the corresponding MODIS data are partially cloudy and potentially underestimate the actual inundation fraction. As a result, it is likely that those pairs will be isolated, used in the retrievals and eventually lead to some underestimation in passive microwave retrievals.

As previously mentioned, the inter-annual climatology of the Mekong delta is highly affected by two tropical monsoons that characterize the seasonal patterns of precipitation, river stages, and water levels (Delgado, et al., 2012). To better understand whether the results of our retrieval algorithm follow the regional climatology, the monthly percentage of inundated area over the Mekong delta is calculated and shown against the monthly water level data in Fig. 8(a). The monthly water level data are obtained by averaging over all 11 stations shown in Fig. 1. The specific goal is to compare the monthly variability of the algorithm outputs with the MWP products and investigate whether they are consistent with the regional variations of the surface water level (river stage), which is considered as surrogate for the extent of inundation. It should be acknowledged that this approach is not a direct validation; however, it can provide insight into the performance and climatological consistency of the proposed model as the surface water level data are positively correlated with the extent of the inundated surfaces.

The seasonal variations in the monthly percentage of the total inundated surfaces from the proposed model follow the trend of monthly water level data better than the standard MWP products (Fig 8(a)). We can see that during the wet months of June-to-November the MWP data report much less inundated area than the outputs of the proposed algorithm, whereas this pattern is reversed during the dry months of January-to-March. As previously noted, we suspect that the differences in the wet season are due to the large portion of missing data in the MWP products because of the high cloud coverage in the rainy season (Fig 8(b)). For quantitative comparison of the outputs of the algorithm with MWP, a Euclidean distance between normalized version of the algorithm outputs and water level data is calculated and compared with its MWP counterpart. The Euclidean distance between water level and the retrieved inundation from ascending and descending orbits is 3.46 and 3.56, respectively, while this distance for MWP and water level data is about 7.89, which is more than twice the distances calculated from the microwave

retrieval results. This indicates the superior performance of the proposed inundation fraction retrievals as compared to the MWP products, chiefly because of its all-sky skills during the rain dominant seasons.

When is compared to MODIS-MWP, the inundated area obtained by the retrieval algorithm in the dry months (Fig 8(a)) shows some underestimation. One reason for this underestimation is the general limitation of the empirical Bayesian estimation method regarding the extreme events (see Coles and Powell, 1996, and the references therein) and we suspect that it is not just limited to the months of January-to-March but it affects the retrievals at other months to a lesser extent, as well. This limitation arises by the sample scarcity of large flooding scenarios during the warm months of the year, which probably lead to the underestimation of inundation fractions related to those events by our retrieval algorithm. We expect that by improving the representativeness of the dataset—especially for extreme events in the summer months, this shortcoming can be significantly improved.

A closer look at Fig. 8(a) also reveals slightly larger inundated surfaces in each month for the ascending (evening overpasses) compared to the descending (morning overpasses) tracks. This small difference between the ascending and descending retrievals can be attributed to the expected diurnal patterns of the precipitation over the Mekong delta. Indeed, it is well documented (Gupta 2005) that localized convective precipitation events are more likely during the evening, which can increase the extent of the inundated areas. To further assess the proposed algorithm performance at a daily scale, we compare the dependence of the total area of ascending daily inundation fractions of the algorithmic outputs with the average daily water level data, using Spearman's rank correlation coefficient. The rationale is that a stronger rank correlation of an inundation product with the water level data implies an improved retrieval. The correlation coefficient between the daily water level of the rivers and the total inundated surfaces of the Mekong delta is equal to 0.22, which drops to -0.38 for the MWP products. To go beyond a rank correlation, we also examined the dependence structures across different ranges of inundation and water level quantiles using an empirical Copula (see Appendix 1).

Copulas provide an effective nonparametric way for simple representation of multivariate joint distributions of high-dimensional random variables to describe their dependence structure. When dependence of two random variables increases, their bivariate "L-shaped" cumulative Copulas tend more to the origin. In Fig. 9, the axes show the marginal quantiles of daily inundation fractions versus those of water level elevations and the contours trace the cumulative Copulas. To characterize the dependence of water level and inundation as a function of topography, we divided the study area into two sub-regions covering the steeper upper parts (above the Phnom Penh gauge in Fig. 1) and the flatter downstream region. The copula analysis for each region was presented separately in Fig. 9. As is evident, the empirical Copula of the total daily inundation fraction from the proposed algorithm shows higher degree of dependence to the water level, as compared to MWP, especially for the quantiles with less than 0.8 cumulative probability for both upstream and the downstream regions. However, comparing the downstream (Fig. 9(a)) and the upstream (Fig 9(b)) regions, we see an increased dependency of the retrievals with the water

levels in the upstream region. This observation seems to be consistent from a geomorphological point of view. Because, over steeper region of the basin the hill slopes are naturally steeper and any small water variability can give rise to significant water extension of inundated areas. However, over fat floodplains water levels and extent of inundations may not be strongly correlated as small changes of water levels my give rise a large extension of flooded surfaces.

## 5 Conclusions and Future Directions

In this paper, we introduced a methodology to retrieve inundation from space for almost all-sky conditions to reduce the gaps that exist in using satellite data in visible to microwave bands. The key idea of the proposed method was to explore the links between overlapping daily high-resolution observations in the visible and near infrared bands from the MODIS and the lower-resolution passive microwave observations from the Special Sensor Microwave Imager/Sounder (SSMIS) sensor. The developed multi-frequency inundation retrieval algorithm uses the $K$-nearest matching method in conjunction with a sparsity promoting regularization technique. The proposed method demonstrated promising results in resolving the spatial patterns of inundation, compared with the MODIS-MWP data. Over the months with high cloud coverage, the monthly results are consistent with the seasonal dynamics of water level variation, which is controlled by tropical monsoons in the Mekong delta. Analysis also showed that, at a daily time scale, the outputs of the algorithm exhibit stronger dependency with the water level data than the MWP data.

There were three major sources of uncertainty in the proposed retrieval model in this paper. The first one related to the use of the 3-day composite MODIS-MWP data (daily products of MODIS-MWP were avoided due to missing values and cloud blockages), which might have introduced some bias in the daily retrievals due to mismatch of time scales. This source of error can be significantly reduced if the MODIS dictionary is populated with more accurate daily products. The second source of error related to the lack of adequate fully clear-sky samples in our dictionary and therefore the need to define a cloud coverage threshold in order to increase the sample size. Using partially cloudy MODIS data was the main reason for some observed underestimation of inundation fractions, especially in the dry months (Figs. 7 and 8), which can be mitigated by increasing the sample size. The last source of error was more related to the general limitation of the Bayesian estimation method regarding the retrieval of extreme events (see, Coles and Powell, 1996, and references therein). This limitation is due to scarcity of large floods in the dictionary, which can be treated by adding more scenarios of extreme events to the dataset from different geographic locations.

One of the limitations of the proposed algorithm (because of the spatial resolution of microwave data used in this paper) was its lack of information about the spatial patterns of inundation within the 12.5 km pixels. The spatial pattern of the estimated inundation fractions can be further enhanced by using the guidance of a high-resolution topographic data (see Galantowicz, 2002). The database can also expand to include some high-resolution cloud-free imageries from newly launched satellites such

Sentinel-2 that can aid in capturing the high-resolution inundation areas. Finally, expanding the dictionary to include data from the passive microwave channels of the new satellites such as Global Precipitation Mission (GPM) Microwave Imager (GMI) will increase the spatial resolution of the retrievals to approximately 5 km. In this paper, the seasonality and also different land surface classes have not been directly taken into account in the retrieval algorithm. Future research should include the

stratification of the dictionary based on different land surface types and time periods (e.g. seasons).

*Acknowledgments:* This work was supported by the NASA Global Precipitation Measurement Program under grants NNX13AG33G and NNX16AO56G. It was also partially supported by NSF under the Belmont Forum DELTAS project (EAR-1342944) and the LIFE project (EAR-1242458). The MODIS-MWP data over the Mekong delta were kindly provided by Dr.

Dan Slayback from the NASA Goddard Space Flight Center. First author would like to thank Professor Robert Brakenridge for his advice on this research during the AGU Fall Meeting 2015.

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

**Acronyms and Abbreviations**

| | |
|---|---|
| SSMIS | Special Sensor Microwave Imager and Sounder |
| SSM/I | Special Sensor Microwave Imager |
| DMSP | Defense Meteorological Satellite Program |
| MSS | Multispectral Scanner System |
| VNIR | Visible to near infrared |
| MODIS | Moderate Resolution Imaging Spectroradiometer |
| NIR | Near infrared |
| MIR | Mid-infrared |
| PMW | Passive microwaves |
| ESMR | Electrically Scanning Microwave Radiometer |
| SMMR | Multi-frequency Microwave Radiometer |
| BWI | Wetness Index |
| WSF | Water Surface Fraction |
| AMSR-E | Advanced Microwave Scanning Radiometer - Earth Observing System |
| NRT | Near Real-Time |
| NSIDC | National Snow and Ice Data Center |
| DFO | Dartmouth Flood Observatory |
| MODIS-MWP | MODIS Near Real-Time (NRT) Water Product |
| CDF | Cumulative probability function |
| $M$ | Number of vectors of microwave brightness temperatures $\mathbf{B}$ |
| $k$ | Number of nearest neigbirs |
| $\mathbf{B}$ | Brightness temperature dictionary |
| $f$ | Inundation fraction |
| $\mathbf{F}$ | Inundation dictionary |
| $\mathbf{b}_{obs}$ | Observed vector of brightness temperature |
| $K$ | Number of nearest neighbors |
| $\mathbf{B}_s$ | Sub-dictionary of $\mathbf{B}$ |
| $\mathbf{F}_s$ | Sub-dictionary of $\mathbf{F}$ |
| $\mathbf{c}$ | Vector of representation coefficients |
| $\hat{f}$ | Estimated inundation fraction |
| $\mathbf{W}$ | Weight matrix |
| n | Number of frequency channels |
| $p$ | Detection probability $\in (0,1)$ |
| $\ell_1$ & $\ell_2$ | Regularizations norms |
| $\lambda_1$ & $\lambda_2$ | Regularization parameters |

## Appendix 1: Copula

Let $X_1$ and $X_2$ denote two random variables with marginal cumulative distributions $F_1(x_1) \equiv P[X_1 \leq x_1]$ and $F_2(x_2) \equiv P[X_2 \leq x_2]$ with the cumulative joint distribution function $F(x_1, x_2) \equiv P[X_1 \leq x_1, X_2 \leq x_2]$. According to the Sklar's theorem (Nelsen, 1999), the cumulative joint distribution $F(x_1, x_2)$ of $X_1$ and $X_2$ is equal to the cumulative joint

5 distribution function $C(u_1, u_2)$ of the quantiles $u_1 = F_1(x_1)$ and $u_2 = F_2(x_2)$ by:

$$
\begin{aligned}
F(x_1, x_2) &= P[X_1 \leq x_1, X_2 \leq x_2] \\
&= P[X_1 \leq F_1^{-1}(u_1), X_2 \leq F_2^{-1}(u_2)] \\
&\equiv C[U_1 \leq u_1, U_2 \leq u_2] \\
&= C(u_1, u_2)
\end{aligned}
\tag{1}
$$

where $C(u_1, u_2)$, is the cumulative Copula with uniform marginal random variables $F_1(x_1)$ and $F_2(x_2)$ on the interval [0, 1]. The multivariate density function $f(x_1, x_2)$, if exists, can be calculated by taking the derivative of $C$ and $F$ which results in the following:

$$
\begin{aligned}
f(x_1, x_2) &= c(u_1, u_2).f(x_1).f(x_2) \\
&= c(F(X_1), F(X_2)).f(x_1).f(x_2)
\end{aligned}
\tag{2}
$$

It shows the Copula density function $c(u_1, u_2)$ separates the joint distribution function $f(x_1, x_2)$ from its marginal probability distribution functions $f(x_1)$ and $f(x_2)$; therefore, it can captures the probabilistic dependence between two random variables $x_1$ and $x_2$ by quantifying the strength of the relationship between their corresponding quantiles.

**Figures**

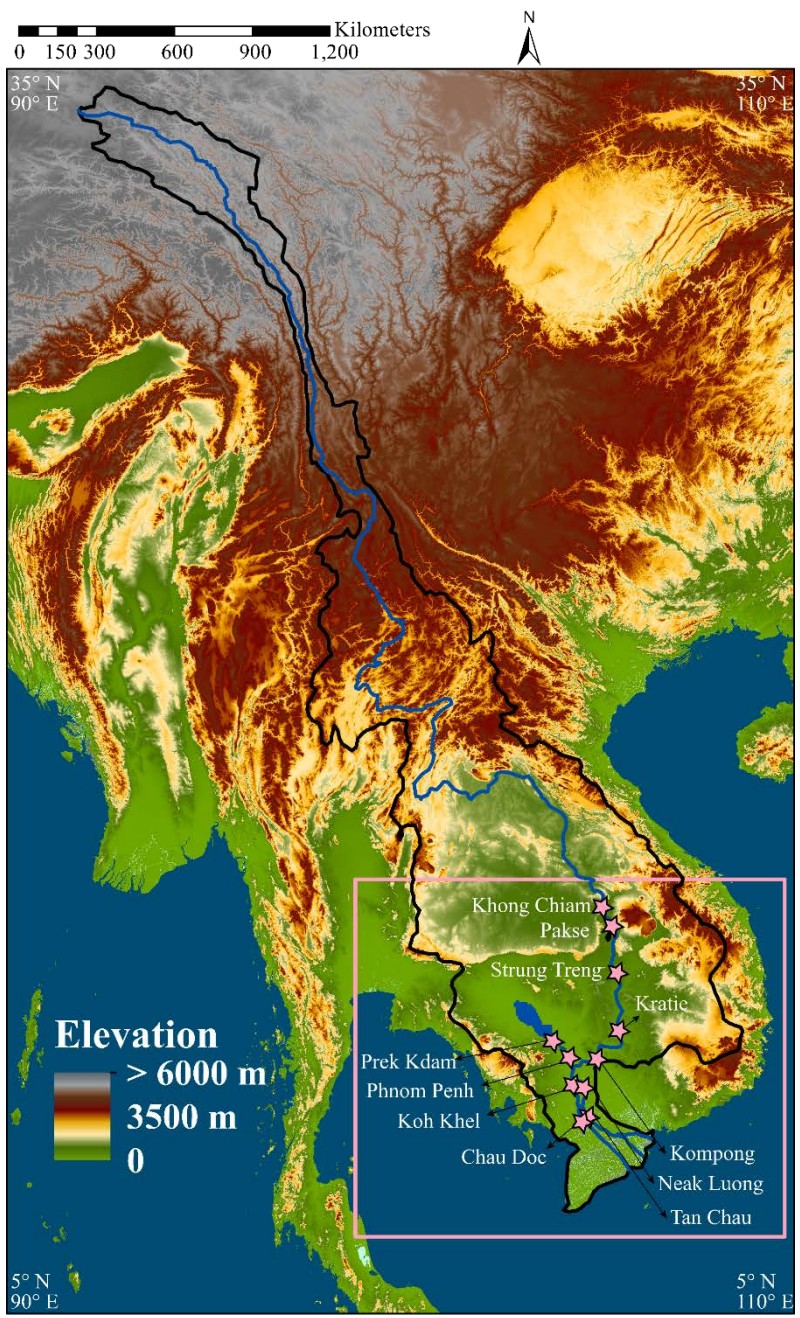

**Figure 1. Map and digital elevation of the Mekong river basin (area=795,000 km2) and its delta. The study area is delineated by a pink rectangle. The 11 stations (from Mekong River Commission) that monitor the water level are also marked by pink stars.**

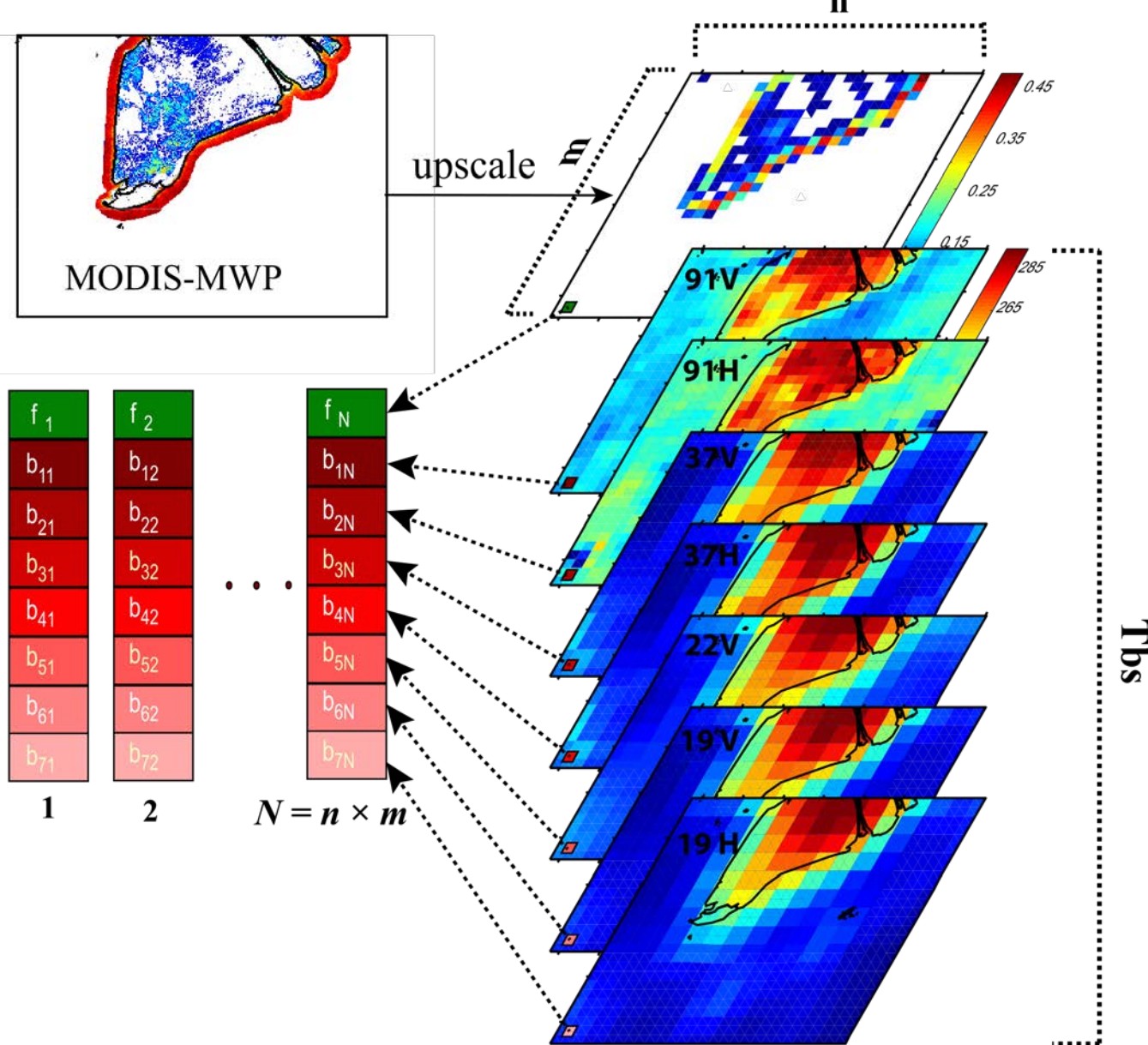

**Figure 2. A Schematic showing construction steps of the *a priori* dataset for dictionaries. The top slab is the upscaled MODIS-MWP and the other slabs are the brightness temperature data at seven frequency bands. Each vector on the left is created by stacking a pixel-level information of the multi-frequency brightness temperatures by the SSMIS radiometer and their corresponding inundation fractions from the MWP product at 12.5 km resolution. This process is repeated for each orbit to generate a large number of vectors and form separate dictionaries for ascending and descending orbits using all satellite overpasses in 5 years from 2010 to 2014. $N = n \times m$ is the number of collected vectors for one day in a year. The same process is conducted for each day in 5 years (2010-2014) to create the dictionaries with $M = \sum_{i=1}^{5 \times 365} N_i$ vectors.**

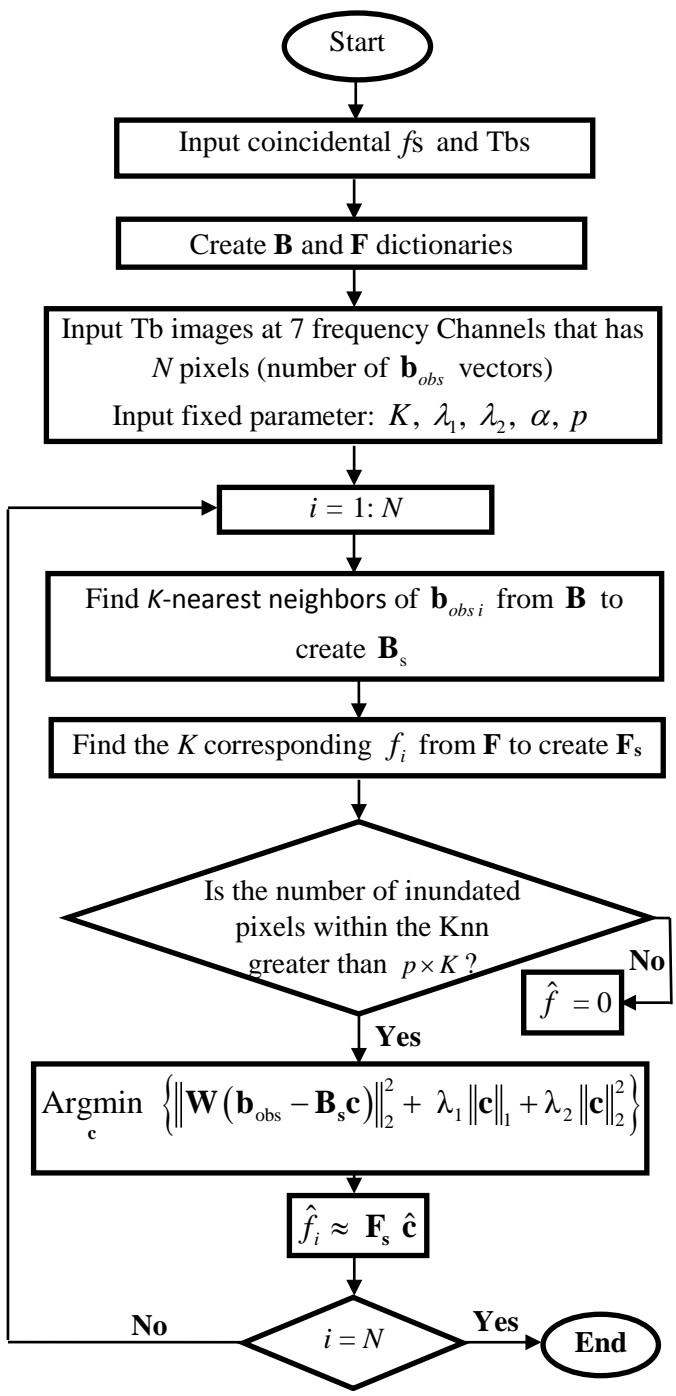

**Figure 3. Flowchart of the inundation retrieval algorithm for *N* pixels in each orbit. See text for definitions of the notations and detailed explanation.**

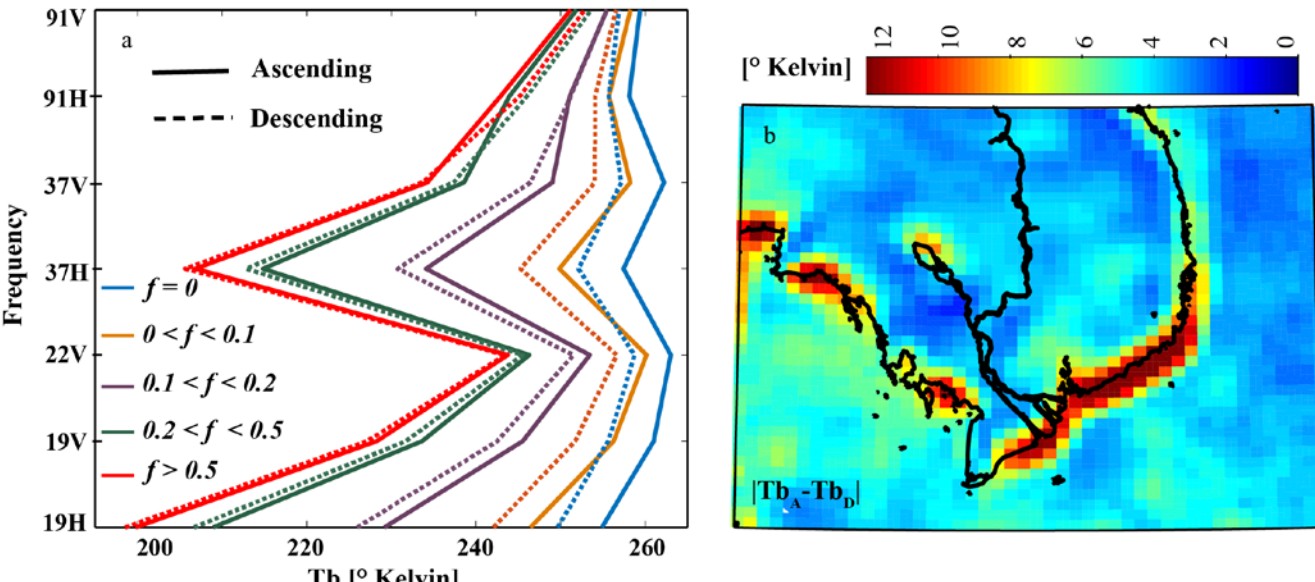

**Figure 4. (a) The systematic difference between passive microwave observations from the ascending (solid lines) and descending orbits (broken lines) as a function of five different sub-pixel intervals of inundation fractions. (b) July-to-Dec daily average of absolute differences between the ascending ($Tb_A$) and descending ($Tb_D$) brightness temperatures at vertically polarized 19 GHz channel. The values of $|Tb_A - Tb_D|$ mainly capture the coastal regions with significant variability in their surface emissivity values due frequent diurnal tidal effects.**

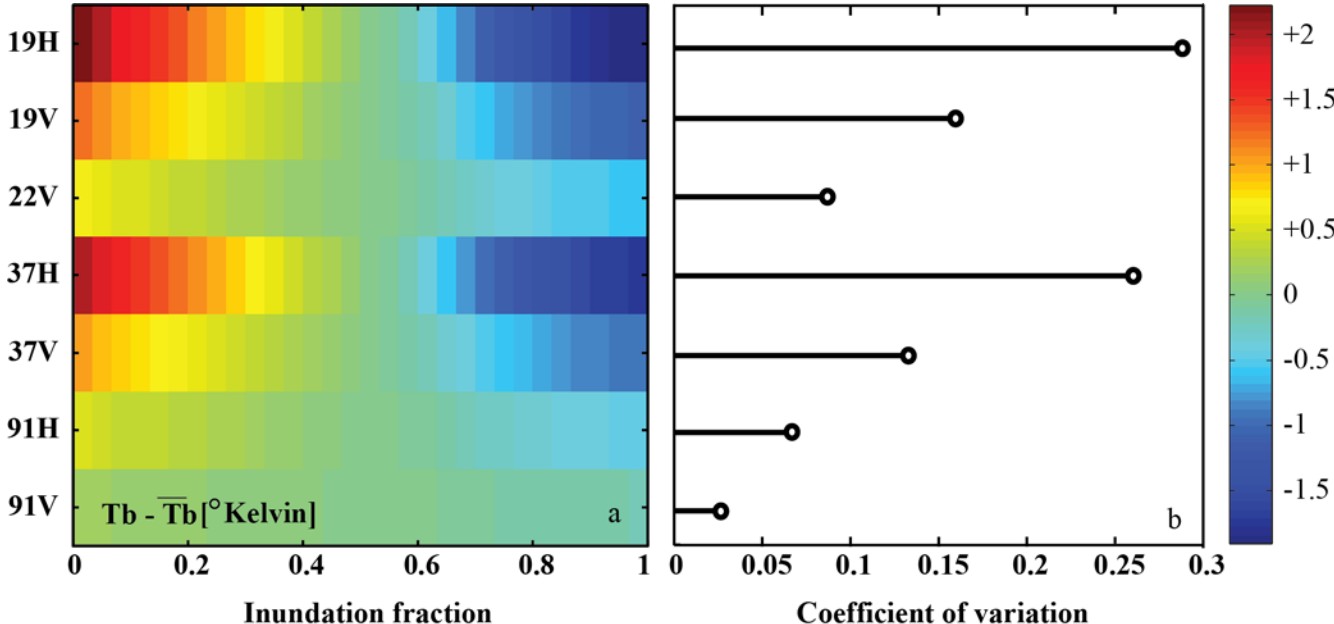

**Figure 5. The normalized coefficients of variation (right panel) of the brightness temperatures (Tb) (left panel) averaged over the entire dataset for different intervals of inundation fractions. Here, $\overline{\overline{Tb}}$ denotes the average of brightness temperatures over the inundation fractions. The coefficients of variation of each channel are used to determine the channel weights for the retrieval algorithm. Channels 19 H GHz and 37 V GHz are the most responsive channels to the variability of inundation fraction and are given higher weights.**

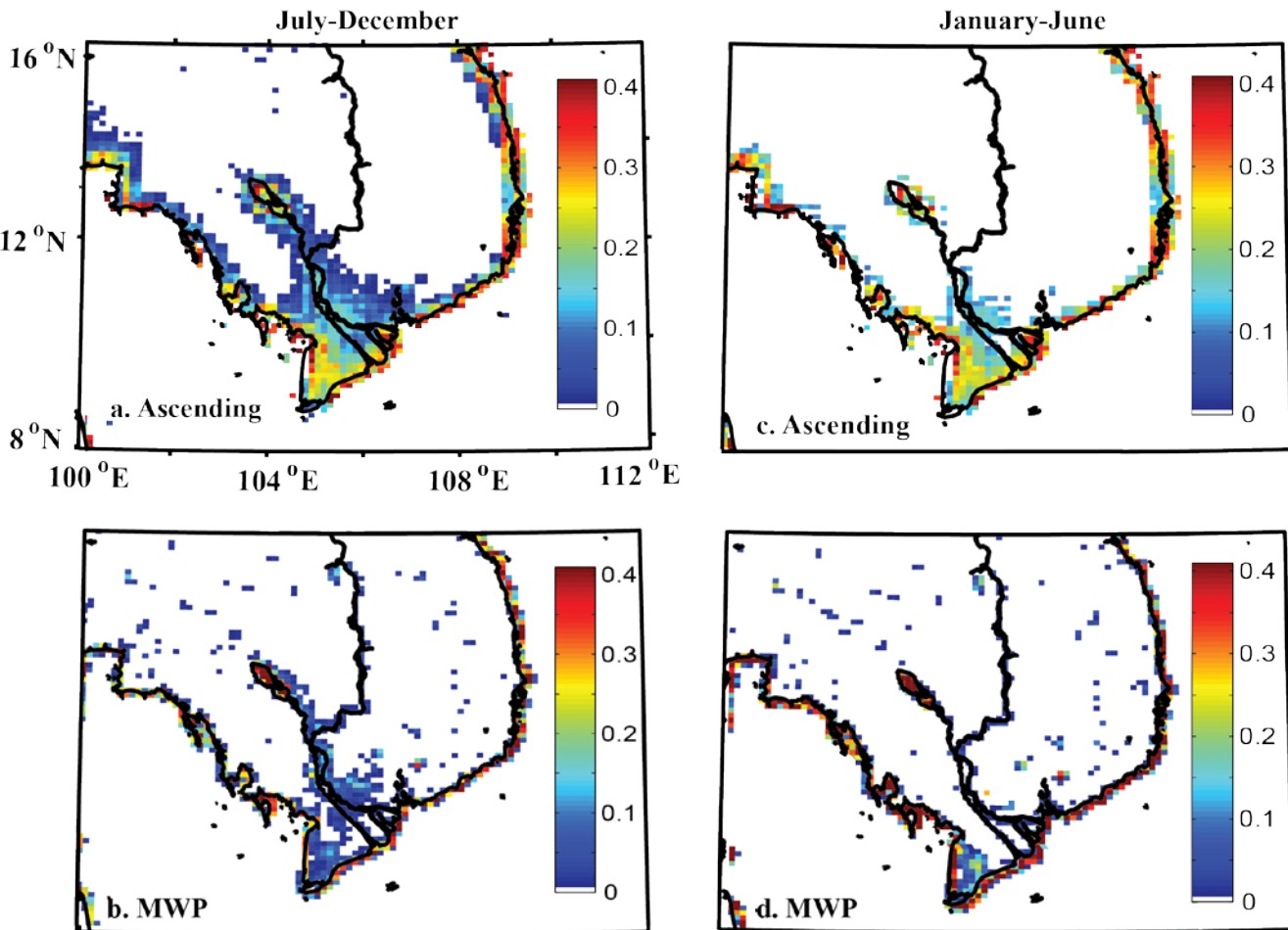

**Figure 6. Inundated map of the Mekong delta in the wet (July-December) and dry (January-June) seasons for the ascending orbits. The results of the proposed retrieval algorithm are presented using the ascending dictionary (top row) against the upscaled MODIS Near Real-Time (NRT) Water Product (MWP) data (bottom row). Overall, a good agreement is observed with some overestimation of inundated areas by the proposed algorithm compared to MODIS-MWP data around the river banks.**

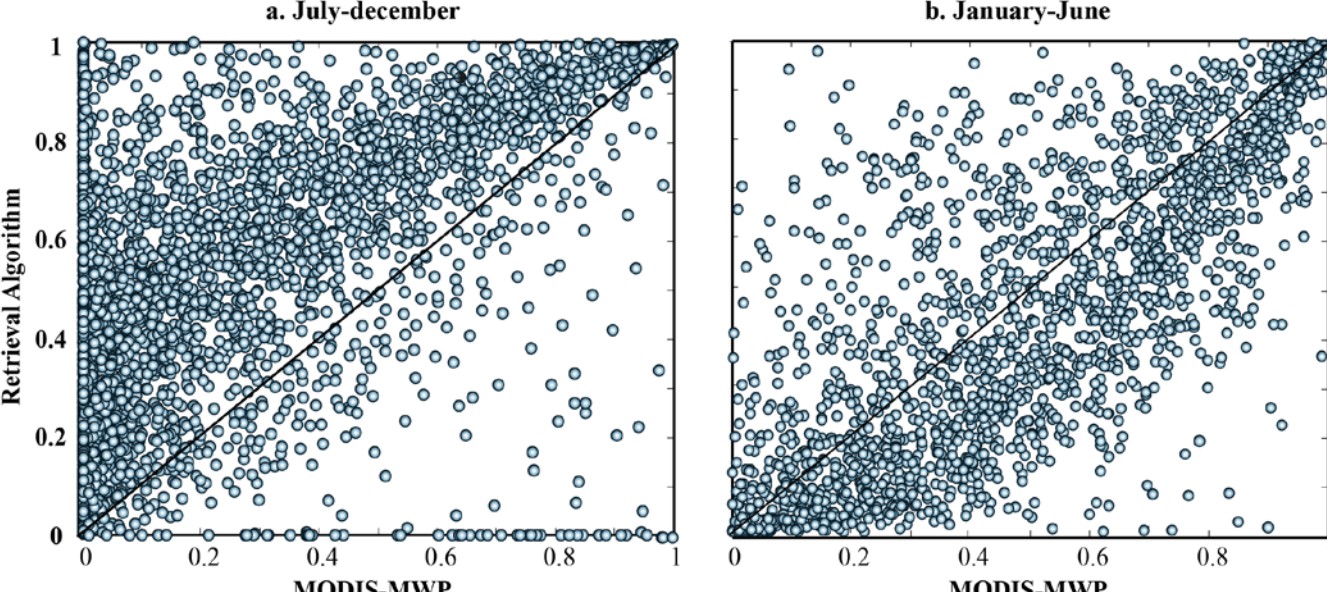

**Figure 7. Scatterplots of daily inundation fractions (*f*) from the retrieval algorithm against those from MODIS-MWP in wet (a) and dry seasons (b) shown in Fig. 6. The scatterplots demonstrate larger inundation fractions from the retrieval algorithm in July-to-December (a) compared to MODIS-MWP data. However, in January-to-June, when there are fewer clouds, the inundation fractions from the proposed algorithm are more correlated with the MODIS-MWP data, with only a slight underestimation of their variability.**

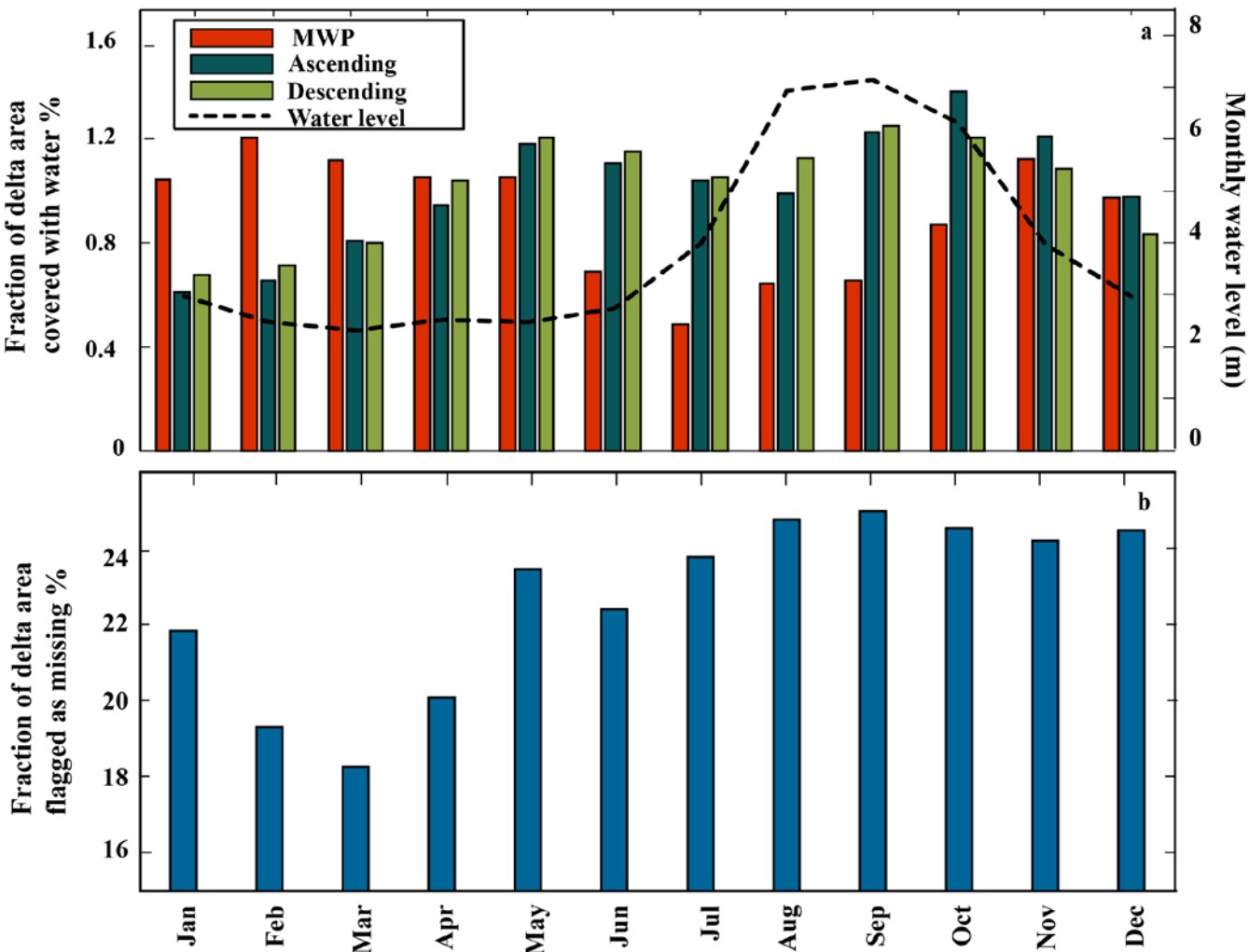

**Figure 8. The monthly inundated areas of the Mekong delta calculated from the proposed retrieval algorithm and MODIS Near Real-Time (NRT) Water Product (MWP) data in comparison with ground based monthly water level data. (a) Comparison of the total inundated surface of the Mekong delta from MWP products and the retrieval algorithm from ascending and descending dictionaries. From visual inspection, it is obvious that the retrieval algorithm can better follow variations of the water levels compared to MWP. More inundation over dry season is reported by MWP products than the wet season, which contradicts the causality between rivers' stages and the extent of inundated areas. (b) The total fraction of land surface areas that are labeled as missing in MWP product because of atmospheric contaminations. The larger deviations of the MWP products from water level data during the wet months might be attributed to the larger percentage of missing values.**

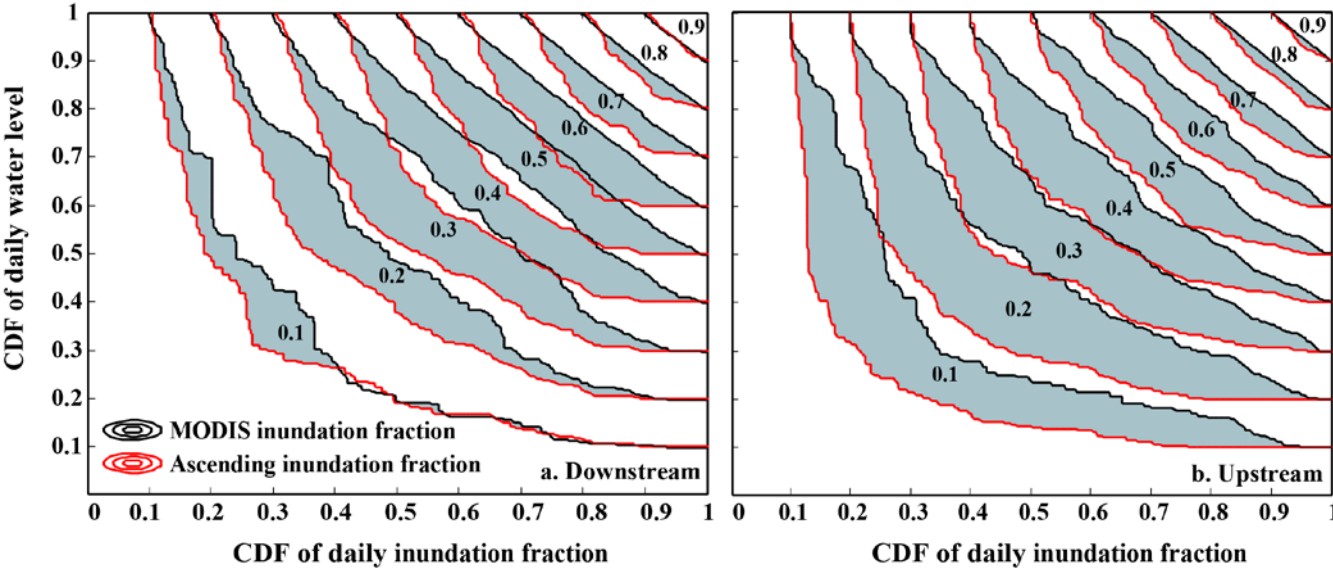

**Figure 9.** The empirical Copula (joint probability distribution of quantiles) of the average daily water level and total daily inundated areas from the proposed retrieval algorithm (red curves) and MODIS-MWP data (black curves) for 2015. These plots indicate that our products have stronger dependence to water levels than the MWP products (more L-shaped curves) for both the downstream (a) and upstream (b) regions of the Mekong Delta. The shaded areas (which quantify the difference between the degree of dependence of our products and the MWP products to the daily water levels) are larger in the upstream region, indicating an enhanced performance of the proposed algorithm to retrieve inundation fraction where potential inundation areas are better defined due to topography, e.g. around major riverbanks.

