# Peer review of "A Multi-sensor Data-driven methodology for all-sky Passive Microwave Inundation Retrieval"

_Hydrology and Earth System Sciences, 2016_

## Referee Comment (RC1) · Anonymous Referee #1 · 1 Dec 2016

This paper presents a new retrieval algorithm for estimating the fraction of water within a passive microwave pixel by using an archive of brightness temperature-MODIS water relationships developed from near-coincident imagery. The manuscript is well written and covers a topic of interest to the remote sensing/hydrology research community. I believe this manuscript is suitable for publication subject to reviewing the minor suggestions below:

I question the use of a 50% clear-sky in the VNIR data for it to be used in the algorithm. Wouldn't this influence the results (which you suggest in the discussion anyway). The Mekong region is very cloudy during the flood season. Would it be better to increase your clear-sky % to higher (which will reduce your number of observations in your

[Figure]

dictionary, but it may improve the results)? Have you already tested this? I suggest including a short discussion on this in the paper.

You use the 3-day composite MWP products to reduce cloud cover. However won't this reduce the accuracy of the near-coincident relationship between the brightness temperature and MODIS water product – especially when you are also looking at the sub-daily diurnal effects? I know you also average the brightness temperature over 3 days as well, however I think it would be worthwhile discussing this possible affect in the manuscript.

Minor corrections: - Page 1, line 11 – should it be 'shortwave infrared'?

- Page 3, line 12 – change 'location' to 'locations'.

- Page 3, line 24 – I suggest changing …'with overlapping in spatial and time…' to 'which overlap in the spatial and time …..

- Page 4, line 15 – change 'form' to 'from'.

- Page 4, line 20 – change '…fraction at resolution…' to '…fraction at spatial resolution…'.

- Page 5, line 23 – change 'form' to 'from'.

- Page 6, line 11 – I suggest changing from first person (i.e. 'let us'). - Page 10, line 2 – change 'that' to 'than'.

- Page 10, line 3 – change 'reverse' to 'reversed'.

- Page 11, line 19 – change 'form' to 'from'.

Figure 2 – should it be 'downscale' rather than 'upscale' since you are reducing the spatial resolution?

Figure 4 caption – I assume the inundation intervals are 'f'. I suggest changing the caption to '…five different inundation fraction intervals (f). …'.

Figure 6 – typing error in the figure 'July-Descember' should be 'July-December'.

[Figure]

---

## Referee Comment (RC2) · Anonymous Referee #2 · 2 Jan 2017

The manuscript has successfully demonstrated a new algorithm for estimating sub-pixel inundated fractions under all weather conditions. By pairing SSMIS multi-frequency observations with MODIS based flood area values during the training period, a weight matrix is identified such that the inundated fraction of a given pixel can be estimated solely from the multi-frequency SSMIS observations over the K-nearest neighbors. This research is built upon traditional wetland/flood mapping approaches that use either passive microwave or VIS/IR alone. The improved spatial and temporal resolutions will contribute to flood monitoring skills during monsoonal seasons. The manuscript is overall well written, but a few areas need further clarification and/or improvement.

Detailed comments:

1) I strongly recommend improving the description of the retrieval algorithm (Section 3).

a) The most important component missing in this section is information about estimating inundated fraction solely from passive microwave observations (e.g., for the year 2015, or during the monsoon season). As shown in the flowchart (Figure 3), the last step is to calculate the inundated fraction using Eq (2), where the coefficient matrix c is optimized from microwave observations (Eq. (4)) and the corresponding inundation fraction (in Fs) is from MODIS (i.e., MWP). How does this work in cases where the Fs value from MODIS is unavailable? I assume the 'dictionaries' (from 2010-2014) are used, but I couldn't find the relevant text?

b) The number of vectors in matrix B needs to be consistent throughout the manuscript. The dimension is n-by-M according to Line 12 on Page 6, where n is the number of frequency channels (i.e. 7) and M is the number of vectors. However, according to Figure 2 N is the number of vectors (and N=n×m), which is confusing. Similarly, it is unclear if the M vectors (Page 6, Line 11) refer to microwave observations in both time and space—or just in space? Assume the domain contains 10 rows and 20 columns, and there are microwave observations for over 300 hundred time steps. Does this mean that M=10×20 (as indicated in Figure 2), or that M=10×20×300 (which is more likely)?

c) Because the K-nearest neighbor search is essential for this study, a bit more information on this process will be helpful. This also relates to the above comment (1b)—will the K- neighbors be selected from one time step, or from multiple observations that occur during different time steps? Since the K- neighbors have a better chance of being geographically close to the pixel of interest (and are from the same time step), will the random selection of 2×106 pairs of brightness temperature and inundation fraction make the Knn less representative?

d) Parameters $\lambda 1$ and $\lambda 2$ in Eq. (4) are not defined until at the end of Section 3. The selection of $\lambda$ and $\alpha$ are made through "cross validation studies", which are not explained.

e) In Figure 3, there are a few constants that are never explained and never provided with values in the manuscripts (such as K, kP, and p).

2) In Section 4, the validation conducted using the probability of "hit" and "false alarm" should be compared between the dry season and wet season. This will help to better understand the results. For instance, there are much fewer missing data points from the MWP during the dry season than during the wet season. Does this mean that there will be a smaller probability of false alarms accordingly—or can the cloud cover/flag from the MODIS product be used to compare results over the 12.5 km pixels with and without cloud contamination?

3) Figures 7a and 7c indicate an overestimation (as compared to 7b and 7d) in regions close to the rivers, and an underestimation in regions not connected to major rivers. Please consider adding some discussion on this.

4) The highlight of this algorithm is the capability to produce inundated sub-pixel fraction results under all-weather at a daily temporal resolution. Therefore, results and validations which contribute to evaluating these skills are preferred. Specifically, it would be interesting to see 1-2 examples showing the daily results (similar to Fig. 7), and comparisons of the sub-pixel fraction values (e.g. using scatter plots) between the MWP and microwave based estimations.

5) There are a number of reasons contributing to the mismatch between the MWP and microwave based estimations. Something important missed in the discussion is the error associated with the MWP. Some discussion about the uncertainties associated with the results is recommended.

6) Although I agree that the water level and the inundated area are correlated, I don't

think it is the best practice to simply average the water levels from 11 gauges to represent the basin. During a flood event, the water level at an upstream gauge located in a steep valley may increase a lot more (and/or faster) than a downstream gauge. However, the downstream gauge is more representative of the basin's condition.

7) A few minor issues:

a) Page 9, line 1: Change "problem" to "equation".

b) Page 9, line 7: It should be Fig. 6, not Fig. 7.

c) Fig. 3: If the Tb images are intended for all years (see comment 1c), please revise the figure accordingly.

d) Fig. 4b: This figure needs units.

e) Fig. 5: Should the word "weights" be removed from the top of the right panel?

f) In some of the figures, the panels are denoted by a, b, c, etc.—but not in all cases. Please be consistent.

---

## Author Comment (AC1) · 22 Jan 2017

This paper presents a new retrieval algorithm for estimating the fraction of water within a passive microwave pixel by using an archive of brightness temperature-MODIS water relationships developed from near-coincident imagery. The manuscript is well written and covers a topic of interest to the remote sensing/hydrology research community. I believe this manuscript is suitable for publication subject to reviewing the minor suggestions below:

We thank the reviewer for the positive view and constructive comments. We took all the comments seriously and incorporated them into a revised manuscript. Detailed replies follow.

I question the use of a 50% clear-sky in the VNIR data for it to be used in the algorithm. Wouldn't this influence the results (which you suggest in the discussion anyway). The Mekong region is very cloudy during the flood season. Would it be better to increase your clear-sky % to higher (which will reduce your number of observations in your dictionary, but it may improve the results)? Have you already tested this? I suggest including a short discussion on this in the paper.

The 50% threshold was selected because the MODIS sensor has a much higher resolution than the footprint of SSMI/S. Therefore, because of limited number of cloud free samples over the Mekong, we need to set a threshold to keep a certain number of SMMI/S cloudy pixels and make sure that the dictionary will not be undersampled. We believe that pixels with a small fraction of cloud cover still contain useful information about surface inundation as microwave observations are not blocked by the presence of clouds. For choosing the threshold we conducted some preliminary analyzes by randomly separating 10% of the samples from the dictionary and estimating the inundation fraction for them based on different thresholds. The analysis showed the 50% threshold, as a fair probability choice, results in minimum values of potential biases. We certainly agree that this is an important issue, which may affect the results to some extent; however, that is the best choice given the available size of the sample space. For future studies, we aim to extend the sample space to other geographical regions to be able collect enough samples with minimal or no cloud cover. We will address this issue in the revised manuscript.

You use the 3-day composite MWP products to reduce cloud cover. However, won't this reduce the accuracy of the near-coincident relationship between the brightness temperature and MODIS water product – especially when you are also looking at the sub-daily diurnal effects? I know you also average the brightness temperature over 3 days as well, however I think it would be worthwhile discussing this possible affect in the manuscript.

We thank you for this comment and acknowledge the existing uncertainties with respect to populating the dictionaries with a 3-day composite MODIS-MWP inundation dataset. We agree that this choice might affect the accuracy of the retrievals. Between the daily and a three-day averaged MODIS product, we decided to use the latter because of its lower retrieval uncertainty. MODIS-MWP daily data are very uncertain because of the shadows of terrains and clouds (Nigro et al. 2014). Typically, there are numerous missing pixels in daily products, which reduce the sample size dramatically. These errors are significantly reduced in the 3-day composite products, as it is less likely for clouds (and their shadows) to stay at the same spot for three days (Nigro et al. 2014). Because the retrievals of the presented method use a weighted average representation of the dictionary atoms, we believe that less uncertain atoms (obtained based on a 3-day MWP dataset) will provide improved estimates of inundation—compared to the more uncertain daily samples. However, a more detailed investigation is certainly needed in future studies. A brief discussion on this source of uncertainty will be added to the revised manuscript.

Minor corrections:

Once again, we would like to thank the reviewer for the provided comments. Please see our detailed response as follows:

We appreciate the reviewer's attention to minor details and the comments she/he has provided. We have incorporated all of these suggestions in our revised manuscript.

- Page 1, line 11 – should it be 'shortwave infrared'? It has been changed to shortwave infrared.

- Page 3, line 12 – change 'location' to 'locations'. It has been changed.

- Page 3, line 24 – I suggest changing…'with overlapping in spatial and time…' to 'which overlap in the spatial and time…'. Revised accordingly.

- Page 4, line 15 – change 'form' to 'from'. Fixed

- Page 4, line 20 – change '…fraction at resolution…' to '…fraction at spatial resolution…'. Revised accordingly.

- Page 5, line 23 – change 'form' to 'from'. It has been changed, thanks.

- Page 6, line 11 – I suggest changing from first person (i.e. 'let us'). - Page 10, line 2. It has been revised.

– change 'that' to 'than'. It has been changed.

- Page 10, line 3 – change 'reverse' to 'reversed'. Fixed.

- Page 11, line 19 – change 'form' to 'from'. It has been corrected. Thanks.

Figure 2 – should it be 'downscale' rather than 'upscale' since you are reducing the spatial resolution?

In earth science, we typically use upscaling when we increase the scale and decrease resolution. We are aware that in different disciplines, this terminology might be used reversely while we associate upscaling with higher resolution.

Figure 4 caption – I assume the inundation intervals are '$f$'. I suggest changing the caption to '…five different inundation fraction intervals ($f$)…'. Incorporated.

Figure 6 – typing error in the figure 'July-Descember' should be 'July-December'. Incorporated.

Reference:

Nigro, J., Slayback, D., Policelli, F. and Brakenridge, G. R.: NASA / DFO MODIS Near Real-Time (NRT) Global Flood Mapping Product Evaluation of Flood and Permanent Water Detection, 2014.

---

## Author Comment (AC2) · 22 Jan 2017

The manuscript has successfully demonstrated a new algorithm for estimating subpixel inundated fractions under all weather conditions. By pairing SSMIS multifrequency observations with MODIS-based flood area values during the training period, a weight matrix is identified such that the inundated fraction of a given pixel can be estimated solely from the multi-frequency SSMIS observations over the K-nearest neighbors. This research is built upon traditional wetland/flood mapping approaches that use either passive microwave or VIS/IR alone. The improved spatial and temporal resolutions will contribute to flood monitoring skills during monsoonal seasons. The manuscript is overall well written, but a few areas need further clarification and/or improvement.

We thank the referee for the valuable insights and comments. We will incorporate the comments into a thorough revision of the manuscript. Detailed replies are provided below.

Detailed comments:

1) I strongly recommend improving the description of the retrieval algorithm (Section 3).

New descriptions will be added to the manuscript based on the below comments.

a) The most important component missing in this section is information about estimating inundated fraction solely from passive microwave observations (e.g., for the year 2015, or during the monsoon season). As shown in the flowchart (Figure 3), the last step is to calculate the inundated fraction using Eq (2), where the coefficient matrix c is optimized from microwave observations (Eq. (4)) and the corresponding inundation fraction (in Fs) is from MODIS (i.e., MWP). How does this work in cases where the Fs value from MODIS is unavailable? I assume the 'dictionaries' (from 2010-2014) are used, but I couldn't find the relevant text?

 $F_s$  is known and represents the inundation dictionary for which each column is attached to the corresponding SSMI/S vector of brightness temperatures  $B_s$ . These two dictionaries are collected using 5 years (2010-2014) of near

coincident SSMI/S brightness temperatures and MODIS-MWP inundation fraction. Please see lines 13-15 on page 6 that explains this concern. For each vector,  $b_i$  in the dictionary of brightness temperatures there is an inundation fraction  $f_i$  from MODIS-MWP. The collection of these pairs from historical observations forms the two dictionaries. We will provide more explanations in the manuscript for further clarification is this issue.

b) The number of vectors in matrix B needs to be consistent throughout the manuscript. The dimension is n-by-M according to Line 12 on Page 6, where n is the number of frequency channels (i.e. 7) and M is the number of vectors. However, according to Figure 2 N is the number of vectors (and N=n×m), which is confusing. Similarly, it is unclear if the M vectors (Page 6, Line 11) refer to microwave observations in both time and space  $A^T$  or just in space? Assume the domain contains 10 rows and 20 columns, and there are microwave observations for over 300 hundred time steps. Does this mean that M=10×20 (as indicated in Figure 2), or that M=10×20×300 (which is more likely)?

We believe that the notation is correct. Capital letters represent the number of vectors. M is the number of total vectors of brightness temperatures collected for all involved orbits in five years of data. However, N

---

## Author Response (AR1)

This paper presents a new retrieval algorithm for estimating the fraction of water within a passive microwave pixel by using an archive of brightness temperature-MODIS water relationships developed from near-coincident imagery. The manuscript is well written and covers a topic of interest to the remote sensing/hydrology research community. I believe this manuscript is suitable for publication subject to reviewing the minor suggestions below:

We thank the reviewer for the positive view and constructive comments. We have addressed all his/her comments in the revised manuscript and provide also detailed replies below.

I question the use of a 50% clear-sky in the VNIR data for it to be used in the algorithm. Wouldn't this influence the results (which you suggest in the discussion anyway). The Mekong region is very cloudy during the flood season. Would it be better to increase your clear-sky % to higher (which will reduce your number of observations in your dictionary, but it may improve the results)? Have you already tested this? I suggest including a short discussion on this in the paper.

The 50% threshold was selected because the MODIS sensor has a much higher resolution than the footprint of SSMI/S. Therefore, because of a limited number of cloud-free samples over the Mekong, we need to set a threshold to keep a certain number of SMMI/S cloudy pixels and make sure that the dictionary will not be undersampled. We believe that pixels with a small fraction of cloud cover still contain useful information about surface inundation as microwave observations are not blocked by the presence of clouds. For choosing the threshold we conducted some preliminary analysis by randomly separating 10% of the samples from the dictionary and estimating their inundation fraction based on different thresholds. The analysis showed that the 50 percent threshold is a good choice as it resulted in bias that was small relative to MODIS-MWP products; for different thresholds, we compared our results with MODIS-MWP over the dry season, which is considered more precise due to less cloud-coverage in that season. For future studies, we aim to extend the sample space to other geographical regions to be able to collect enough samples with minimal or no cloud cover. We have addressed this issue in the revised manuscript on page 8 lines 21–25 and also in the uncertainties and limitations of the algorithm in the "Conclusion and future directions" section in page 13 lines 19–23.

You use the 3-day composite MWP products to reduce cloud cover. However, won't this reduce the accuracy of the near-coincident relationship between the brightness temperature and MODIS water product – especially when you are also looking at the sub-daily diurnal effects? I know you also average the brightness temperature over 3 days as well, however I think it would be worthwhile discussing this possible affect in the manuscript.

We thank you for this comment and acknowledge the existing uncertainties with respect to populating the dictionaries with a 3-day composite MODIS-MWP inundation dataset. We agree that this choice might affect the accuracy of the retrievals. Between the daily and a three-day averaged MODIS product, we decided to use the latter because of its lower retrieval uncertainty. MODIS-MWP daily data are very uncertain because of the shadows of terrains and clouds (Nigro et al. 2014). Typically, there are numerous missing pixels in daily products, which reduce the sample size dramatically. These errors are significantly reduced in the 3-day composite products, as it is less likely for clouds (and their shadows) to stay at the same spot for three days (Nigro et al. 2014). Because the retrievals of the presented method use a weighted average representation of the dictionary atoms, we believe that less uncertain atoms (obtained based on a 3-day MWP dataset) will provide improved estimates of inundation—compared to the more uncertain daily samples. However, a more detailed investigation is certainly needed in future studies. This discussion has been added to page 5 lines 16–22 in the "Study Area and database" session and also in the algorithm limitations and uncertainties in on page 13 lines 15–18.

**Minor corrections:**

Once again, we would like to thank the reviewer for the provided comments. Please see our detailed response as follows:

- Page 1, line 11 should it be 'shortwave infrared'? It has been changed to shortwave infrared.
- Page 3, line 12 change 'location' to 'locations'. It has been changed.

- Page 3, line 24 – I suggest changing...'with overlapping in spatial and time...' to 'which overlap in the spatial and time...'. *Revised accordingly*.

- Page 4, line 15 - change 'form' to 'from'. Fixed

- Page 4, line 20 – change '...fraction at resolution...' to '...fraction at spatial resolution...'. *Revised* accordingly.

- Page 5, line 23 - change 'form' to 'from'. It has been changed, thanks.

Page 6, line 11 – I suggest changing from first person (i.e. 'let us'). - Page 10, line 2. *It has been revised*.
change 'that' to 'than'. *It has been changed*.

- Page 10, line 3 - change 'reverse' to 'reversed'. Fixed

- Page 11, line 19 – change 'form' to 'from'. It has been corrected. Thanks.

Figure 2 – should it be 'downscale' rather than 'upscale' since you are reducing the spatial resolution? In earth science, we typically use upscaling when we increase the scale and decrease resolution. We are aware that in different disciplines, this terminology might be used reversely while we associate upscaling with higher resolution.

Figure 4 caption – I assume the inundation intervals are 'f'. I suggest changing the caption to '...five different inundation fraction intervals (f)...'. *Incorporated*.

Figure 6 – typing error in the figure 'July-Descember' should be 'July-December'. Corrected.

b) The number of vectors in matrix B needs to be consistent throughout the manuscript. The dimension is n-by-M according to Line 12 on Page 6, where n is the number of frequency channels (i.e. 7) and M is the number of vectors. However, according to Figure 2 N is the number of vectors (and N=n×m), which is confusing. Similarly, it is unclear if the M vectors (Page 6, Line 11) refer to microwave observations in both time and space  $\hat{a}A^T$  or just in space? Assume the domain contains 10 rows and 20 columns, and there are microwave observations for over 300 hundred time steps. Does this mean that M=10×20 (as indicated in Figure 2), or that M=10×20×300 (which is more likely)?

 $M=10\times20\times300$  is the correct answer. We believe that the notation is correct. Capital letters represent the number of vectors. M is the total number of brightness temperature vectors collected for all involved orbits in five years of data. However, N

[revised manuscript text omitted]

**Appendix 1: Copula**

5

Let  $X_1$  and  $X_2$  denote two random variables with marginal cumulative distributions  $F_1(x_1) \equiv P[X_1 \leq x_1]$  and  $F_2(x_2) \equiv P[X_2 \leq x_2]$  with the cumulative joint distribution function  $F(x_1, x_2) \equiv P[X_1 \leq x_1, X_2 \leq x_2]$ . According to the Sklar's theorem (Nelsen, 1999), the cumulative joint distribution  $F(x_1, x_2)$  of  $X_1$  and  $X_2$  is equal to the cumulative joint distribution function  $C(u_1, u_2)$  of the quantiles  $u_1 = F_1(x_1)$  and  $u_2 = F_2(x_2)$  by:

$$F(x_{1}, x_{2}) = P[X_{1} \le x_{1}, X_{2} \le x_{2}]$$

=  $P[X_{1} \le F_{1}^{-1}(u_{1}), X_{2} \le F_{2}^{-1}(u_{2})]$
=  $C[U_{1} \le u_{1}, U_{2} \le u_{2}]$
=  $C(u_{1}, u_{2})$
(1)

where  $C(u_1, u_2)$ , is the cumulative Copula with uniform marginal random variables  $F_1(x_1)$  and  $F_2(x_2)$  on the interval [0, 1]. The multivariate density function  $f(x_1, x_2)$ , if exists, can be calculated by taking the derivative of C and F which results in the following:

10
$$\begin{aligned} f(x_1, x_2) &= c(u_1, u_2) . f(x_1) . f(x_2) \\ &= c \big( F(X_1), F(X_2) \big) . f(x_1) . f(x_2) \end{aligned}$$
(2)

It shows the Copula density function  $c(u_1, u_2)$  separates the joint distribution function  $f(x_1, x_2)$  from its marginal probability distribution functions  $f(x_1)$  and  $f(x_2)$ ; therefore, it can capture the probabilistic dependence between two random variables  $x_1$  and  $x_2$  by quantifying the strength of the relationship between their corresponding quantiles.

**Figures**